# Immunotherapy-Related Publications in Colorectal Cancer: A Bibliometric Analysis

**DOI:** 10.3390/healthcare10010075

**Published:** 2021-12-31

**Authors:** Daniel Sur, Cristina Lungulescu, Irina-Ioana Puscariu, Simona Ruxandra Volovat, Madalina Preda, Elena Adriana Mateianu, Cristian Virgil Lungulescu

**Affiliations:** 1Department of Medical Oncology, The Oncology Institute “Prof. Dr. Ion Chiricuţă”, 400015 Cluj-Napoca, Romania; ioanairina98@gmail.com; 2Department of Medical Oncology, University of Medicine and Pharmacy “Iuliu Hatieganu”, 400012 Cluj-Napoca, Romania; 3Doctoral School, University of Medicine and Pharmacy Craiova, 200349 Craiova, Romania; cristina.lungulescu@yahoo.com; 4Department of Medical Oncology, University of Medicine and Pharmacy Grigore T Popa Iasi, 700115 Iasi, Romania; simonavolovat@gmail.com; 5Department of Microbiology, Parasitology and Virology, Faculty of Midwives and Nursing, Carol Davila University of Medicine and Pharmacy, 050474 Bucharest, Romania; madalina.preda@drd.umfcd.ro; 6Institute of Oncology “Prof. Dr. Alexandru Trestioreanu”, Șoseaua Fundeni, 022238 Bucharest, Romania; elena.mateianu@gmail.com; 7Department of Medical Oncology, University of Medicine and Pharmacy Craiova, 200349 Craiova, Romania

**Keywords:** analysis, bibliometric, colorectal cancer, immunotherapy, research, treatment, publications

## Abstract

Patients with microsatellite-instability-high (MSI-H) or mismatched repair-deficient colorectal cancer (CRC) appear to be responsive to checkpoint inhibitors. This study aimed to assess research trends in CRC immunotherapy. Publication patterns of articles covering immunotherapies in CRC in the Web of Science Core Collection database were retrospectively examined using VOS viewer software (version 1.6.16) prior to 25 May 2021. Ultimately, 3977 records were identified that were published between 1975 and 2021, which received a total of 128,681 citations (an average of 32.36 citations per item), with a noticeable rise in 2014. The majority of articles were published in the US (35.8%), China (17.7%), and Germany (9.4%). Publications mainly originated from the Institut National de la Santé Et De La Recherche Medicale Inserm, followed by the University of Texas System and Harvard University; however, Johns Hopkins University received the most citations (18,666 for 69 publications). The Journal of Clinical Oncology issued the most publications (*n* = 146), while the most referenced item (7724 citations) was published in the New England Journal of Medicine in 2012. The most common keywords were associated with tumors (expression and microsatellite instability) or immune system components (t-cells/dendritic cells). The findings demonstrate the scientific community’s interest in the MSI-H subtype of colorectal tumors and how immunotherapy may be employed more successfully to treat metastatic CRC.

## 1. Introduction

Colorectal cancer (CRC) is a common malignancy and one of the primary causes of cancer-related mortality today [1]. Early-stage CRC benefits from curative treatment consisting of either surgery (appropriate tumor resection and lymphadenectomy) or surgery coupled with adjuvant chemotherapy (combination of oxaliplatin and fluoropyrimidine). Metastatic CRC patients are rarely candidates for radical surgery. In such instances, the focus and methods of treatment shift to improve overall survival using chemotherapy, targeted agents, and more recently immunotherapy [2]. Among those breakthroughs, immunotherapy was one of the most promising. Rather than directly targeting cancer cells, immunotherapy helps the host immune system in combating the malignant process [3]. Following the effectiveness of immunotherapy in the treatment of melanoma and renal cell carcinoma, studies on different cancer types were conducted [4,5]. The results for CRC were largely underwhelming with the exception of a rare subtype-deficient mismatch repair/microsatellite instability-high (dMMR/MSI-H) tumors [6]. Currently, the field is considered promising, and research on the subject is ongoing.

Bibliometric’s purpose is to trace the research profiles of different countries, centers, and researchers by quantifying the aspects of the production and dissemination of the knowledge generated over a certain time period and, as a result, offering important analysis of scientific production, behavior, and development in research fields [7]. In many areas of medical studies, including cancer research, a broad range of bibliometric analyses has been employed [8].

This study aimed to evaluate the development and impact of immunotherapy-related articles in CRC. The intended outcome is to gain a better understanding of the current situation and trends of those research studies by analyzing their main characteristics.

## 2. Materials and Methods

This analysis was conducted on 25 May 2021 by using the Web of Science Core Collection database, which has wide coverage with professional article indexing for conducting searches [9]. Inclusion criteria were articles discussing immunotherapies in CRC, where an included article should mention any CRC and immunotherapy keywords in their title, abstract, or keywords, and it discusses CRC immunotherapies. The resulting articles were manually screened, and excluded articles were not related to CRC immunotherapies. Furthermore, corrections, retracted items, or news items were excluded from the analysis. It is worth mentioning that there were no limitations in the searched period and the language of articles.

### 2.1. Search Strategy

The keywords that were used to define CRC immunotherapies include “immunotherapy” OR “anti-CTLA-4” OR “anti-PD-1” OR “anti-PD-L1” OR “Ipilimumab” OR “Tremelimumab “OR “Nivolumab” OR “ Pembrolizumab” OR “ PDR001” OR “Atezolizumab” OR “Durvalumab” OR “ Avelumab”. Moreover, keywords such as “colorectal cancer” OR “cancer of the colon” OR “colorectal malignancy” OR “colorectal tumor” OR “colorectal neoplasm” OR “colon tumor” OR “colon neoplasm” were searched for colorectal cancer. In addition, the Web of Science “Topic” category was specified to be searched for articles containing the above-mentioned keywords in their title, abstract, author keywords, and Keywords Plus.

The quotation marks were employed to define the searched term, and the star “*” operator was utilized to search for a specific stem, regardless of the other part of the word. The resulting search query was as follows: TS = (“immunotherap*” OR “anti-CTLA-4” OR “anti-PD-1” OR “anti-PD-L1” OR “Ipilimumab” OR “Tremelimumab” OR “Nivolumab” OR “Pembrolizumab” OR “pda001” OR “Atezolizumab” OR “Durvalumab” OR “Avelumab”) AND TS = (“colorectal cancer” OR “cancer of the colon” OR “colorectal malignancy” OR “colorectal tumor” OR “colorectal neoplasm” OR “colon tumor” OR “colon neoplasm”).

The searched items were analyzed in the Web of Science platform and were also exported to be analyzed using VOS viewer software as detailed below.

### 2.2. Variables and Analysis

For the included articles, data were extracted regarding the following variables: open access articles, articles published each year, articles affiliated to countries and institutions, top journals publishing articles related to immunotherapy research related to CRC, and top 10 most cited articles related to immunotherapy research related to CRC. Regarding journals, their respective impact factor was extracted from the 2019 journal citation report. A citation report was also created for the included articles, where the total number of citations received and the average number of citations for each item were extracted.

VOS viewer (version 1.6.16) was employed for literature mapping in order to analyze the co-occurrence of keywords, citation, and co-authorship. Moreover, it calculates an indication of how strong an item is connected to the others (an indicator called “Total link strength”) [10]. For each conducted analysis, total documents, citations, and total link strength were reported.

## 3. Results

### 3.1. Web of Science Output

The initial query resulted in a total of 3995 articles. After refining our search and manually excluding non-related articles (Figure 1), a total of 3977 articles were included in our analysis. Furthermore, a total of 2096 (52.7%) were open access articles, and the included articles received a total of 128,681 citations, with an average of 32.36 citations per item.

#### 3.1.1. Years

The first related article was published in 1975 [11], and since then, the number of articles has increased. Since 2014, the yearly number of articles published exceeded 100 articles. Figure 2 shows the number of articles during the last 10 years.

#### 3.1.2. Countries and Institutions

Regarding countries, USA was the top country with 1423 (35.8%) publications, followed by China (*n* = 702; 17.7%) and Germany (*n* = 372; 9.4%). Figure 3 illustrates the top 10 countries regarding immunotherapy research related to CRC. The French-based Institut National De La Sante Et De La Recherche Medicale Inserm was the institution with the highest number of publications, with almost 144 (3.6%) publications, followed by the US-based University of Texas System (*n* = 140; 3.5%) and US-based Harvard University (*n* = 119; 3.0%). Table 1 details the top 10 institutions regarding immunotherapy research related to CRC.

#### 3.1.3. Journals

The journals that published the highest number of papers were the *Journal of Clinical Oncology* (impact factor 32.96) with 146 publications, followed by *Cancer Immunology Immunotherapy* (impact factor 5.44) with 117 publications, and *Oncoimmunology* (Impact factor 5.87) with 95 publications. Table 2 tabulates the top 10 journals regarding the number of papers published related to immunotherapy in CRC according to the Web of Science database.

#### 3.1.4. Top 10 Most Cited Articles

For the top 10 highest cited papers related to immunotherapy research and colorectal cancer, the top-cited paper was published in 2012 in the *New England Journal of Medicine* titled “Safety, Activity, and Immune Correlates of Anti-PD-1 Antibody in Cancer”. Table 3 summarizes the top 10 highly cited papers related to immunotherapy in CRC according to the Web of Science database, all of which were open access articles.

### 3.2. VOS Viewer Analysis (Hotspots)

#### 3.2.1. Keyword Analysis

Keywords provided by the authors of the paper and those that occurred more than 10 times in the WOS core database were enrolled in the final analysis. Of 11,065 keywords, 647 met the threshold and had the greatest total link strength. The keywords that appeared most were “immunotherapy” (1598 occurrences with a total link strength of 12,636) and “colorectal cancer” (1298 occurrences with a total link strength of 9429), and they were excluded from our keyword mapping as they were specifically searched for in the Web of Science database. The top resulting keywords were “expression” (541 occurrences with a total link strength of 4146), “microsatellite instability” (359 occurrences with a total link strength of 3141), and “dendritic cells” (346 occurrences with a total link strength of 2847). Figure 4 shows the bibliometric analysis of keywords in the publications of immunotherapy in CRC showing the co-occurrence of keywords, where the size of nodes indicates the frequency of occurrence.

#### 3.2.2. Citation Analysis for Journals

The journals that received a minimum number of 100 citations were analyzed, and a total of 130 journals met our threshold. *New England Journal of Medicine* had the highest number of citations (19,449 citations for 14 documents with 924 total link strength), followed by *Clinical Cancer Research* (7191 citations for 90 documents with 784 total link strength), and *Journal of Clinical Oncology* (6091 citations for 146 documents with 752 total link strength). Figure 5 depicts bibliometric analysis of the journals in the publications of immunotherapy in CRC, showing journals receiving top citations, where the size of nodes indicates the number of citations received.

#### 3.2.3. Citation Analysis for Institutions

The institutions that had at least 10 publications receiving at least 100 citations were also analyzed, and documents co-authored by authors from more than 25 institutions were excluded. In total, 383 institutions met our thresholds criteria. Johns Hopkins University had the highest number of citations (18,666 citations for 69 documents with 2588 total link strength), followed by Memorial Sloan Kettering Cancer Center (12,025 citations for 93 documents with 1767 total link strength), and Yale University (9858 citations for 17 documents with 718 total link strength). Figure 6 shows the bibliometric analysis of the institutions in the publications of immunotherapy in CRC, showing institutions receiving top citations, where the size of nodes indicates the number of citations received.

#### 3.2.4. Citation Analysis for Countries

The countries that had at least 100 publications receiving at least 1000 citations were analyzed, and documents co-authored by authors from more than 25 countries were excluded. A total of 12 countries met our thresholds criteria. The USA had the highest number of citations (72,060 citations for 1408 documents with 8509 total link strength), followed by France (13,172 citations for 262 documents with 3512 total link strength), and the People’s Republic of China (12,830 citations for 713 documents with 3725 total link strength). Figure 7 demonstrates the bibliometric analysis of the countries in the publications of immunotherapy in CRC showing countries receiving top citations, where the size of nodes indicates the number of citations received.

#### 3.2.5. Co-Authorship Analysis

The authors with at least five publications receiving at least 10 citations were included, and documents co-authored by more than 25 authors were excluded. In total, 394 authors met our criteria. Drew Pardoll had the highest number of citations (4650 citations for eight documents with 30 total link strength), followed by Janis Taube (4394 citations for five documents with 19 total link strength), and Lieping Chen (3779 citations for five documents with 8eighttotal link strength). Figure 8 shows the bibliometric analysis of the authors of publications of immunotherapy in CRC showing authors receiving top citations, where the density indicates the number of citations received.

## 4. Discussion

Health authorities are increasingly recognizing the role of bibliometrics in evidence-based policy and patient care. In every country, knowledge generation in oncology is a vital phase in the care process. By leading and supporting government cancer control strategies, this can have a substantial impact on patient outcome.

This study aimed to analyze immunotherapy research trends for colon cancer. By using bibliometric analysis of the subject, the status and characteristics of relevant publications were explored, including the most cited articles, research institutions, and journals. The objective was to provide ideas and direction to researchers by identifying current trends in this subject. This bibliometric analysis included every related article from the Web of Science Core Collection database until May 2021, rendering a visual and systematic overview of immunotherapy research trends for colon cancer.

Annual publications on the topic have grown gradually since 2012. A significant increase in the number of articles published took place between 2015 and 2016, reflecting the emergence of checkpoint blockade immunotherapy and the subsequent interest it attracted from the research community [21]. For unresectable or metastatic mismatch repair deficient (dMMR) and microsatellite instability-high (MSI-H) CRC, the FDA has approved three immune checkpoint inhibitors (ICIs) targeting PD-1 (pembrolizumab, nivolumab) and CTLA-4 (ipilimumab) in 2017 and 2018, respectively [22,23].

By finding the most common keywords used, we can better understand the direction of academic research and the areas that have been identified as the most promising and deserving of attention. It should come as no surprise that the top keywords found by our analysis are related either to the specific characteristics of the tumor (“expression” and “microsatellite instability”) or to the immune system and its components (“t-cells”/“dendritic cell”). In the study of the effect of immunotherapy in the treatment of CRC, special attention is given to the MSI-H subtype. This type of cancer cell has a high level of microsatellite instability, resulting in a high number of mutations [24]. Such characteristics make it susceptible to checkpoint inhibitors. Although much of the research in this field has concentrated on this variant, there are also attempts to expand the efficacy of immunotherapy to all types of CRC, either by using sensitization to checkpoint inhibition therapy or by using other methods, such as vaccines or CAR-T cell therapy [21].

Adaptive immunotherapy has dominated contemporary cancer immunotherapy research according to an analysis of papers published between 2015 and 2018, regardless of primary tumor site [25]. Subcategory analysis of tumor immunotherapy in the research of adaptive immunity revealed that the most popular subjects were immune checkpoint inhibitors, CAR-T, and cancer vaccines. Dendritic cells were the most studied issues in innate immunity research, possibly because of their capacity to trigger primary immune responses, as well as linking innate and adaptive immunity systems [25]. Using CAR-T cell therapy, the patient’s own T cells are harvested and genetically modified before being reinfused. This approach has shown promise in hematological malignancies [26,27], and it has also been advocated for solid tumors. However, there are no CAR-T cell treatments currently approved for CRC despite several studies that have entered clinical trials as a result of ongoing fundamental research efforts [28].

The USA (*n* = 1423) and China (*n* = 702) are the leading countries by the number of publications, accounting for more than half of all immunotherapy-related articles published. The following positions were occupied by Germany, Italy, and Japan (9%, 8%, and 7%, respectively). The 10 most cited papers in the field have all been published by American journals, suggesting that the USA has remained the frontrunner in immunotherapy research in terms of both quantity and quality over the years. The *New England Journal of Medicine* had the highest number of citations, while *Clinical Cancer Research* and *Journal of Clinical Oncology* ranked number two and three, respectively. Institut National De La Sante Et De La Recherche Medicale Inserm had the most published articles (*n* = 144; 3.621%), followed by the American Universities of Texas System and Harvard (*n* = 140; 3.520% and *n* = 119; 2.992%, respectively). Furthermore, among the first 10 institutions by the number of published papers, other French and German centers also appeared. However, when considering the number of citation articles published by an institution received, Johns Hopkins University leads. The Memorial Sloan Kettering Cancer Center and Yale University place second and third.

According to studies, articles must acquire enough citations two to three years following publication for bibliometric measurements to be valid, as citations accumulate gradually over time [29]. The consequence will be a bias in any ranking of the most highly cited articles published across a specific timeframe in favor of older papers, regardless of the field or journal. Individual authors are subject to the same principle—bibliometric indicators benefit senior writers [30].

Researchers can assess the impact of a series of articles by monitoring their citations and categorizing them further by a single author, organization, or an entire country. Higher citation counts imply a greater level of influence. There are a variety of reasons why authors cite other publications. Motives may include reinforcing a point made in the text, naming sources for methods they employ, clarifying a certain technique, or acknowledging mentors or authorities in the area. Occasionally, authors intend to discuss examples of faulty procedures or outcomes that are deceptive. These variations cannot be addressed by current bibliometric criteria, which tally all citations identically regardless of the citation’s real purpose [30].

The amount of citations a publication receives does not indicate if the study described in that paper had a positive impact on people’s health, which is the collective goal for medical research. The only aspect that can be determined is whether or not the paper was helpful to other writers in the process of producing their own articles [30].

To the best of our knowledge, this is the first bibliometric investigation of immunotherapy in colorectal cancer. The vast majority of publications in the subject of immunotherapy in CRC were covered by data obtained from available sources. The data presented might not represent the entire picture of the immunotherapy research in CRC, because conference papers, medical thesis and patents were not included in document screening. Furthermore, bibliometric indicators have a number of limitations, and caution should be used when interpreting the results of the study.

Our study may suffer from the usual shortcomings of bibliometrics. Our citation analysis was based on the search result of the Web of Science Core Collection database, which might mean that significant publications that were available in other databases, such as Google and Scopus, could have been overlooked. Another small number of articles could have been missed by only searching for keywords in the title, abstract, author keywords, and Keywords Plus. Citation analyses have their own inherent problems, such as reliance on the number of times an article has been cited—an aspect that depends on a multitude of uncontrollable factors.

## 5. Conclusions

The study analyzed global academic publications related to immunotherapy and applied to CRC research quantitatively and qualitatively. There was an increase in the number of publications on the subject worldwide since 2012, and the tendency is toward continued growth in terms of the average volume of published articles. Results also showed the interest the scientific world has for the MSI-H variant of CRC and the way immunotherapy can be used to more effectively treat CRC.

## Figures and Tables

**Figure 1 healthcare-10-00075-f001:**
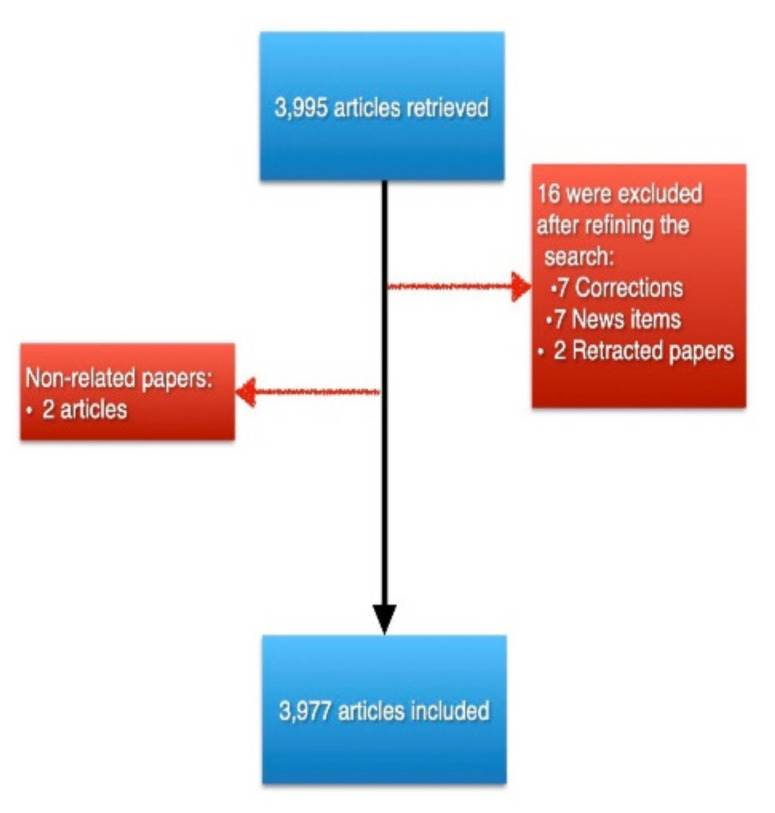
A PRISMA-like graph detailing the number of retrieved records and the resulted final included number after excluding records.

**Figure 2 healthcare-10-00075-f002:**
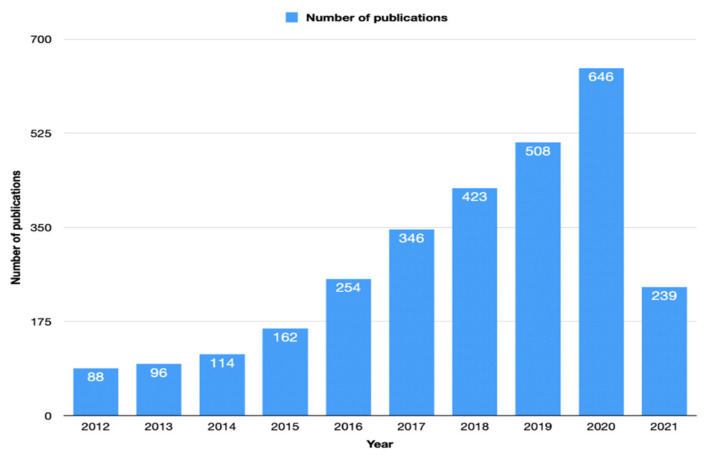
The number of articles during the last 10 years related to immunotherapy in colorectal cancer according to the Web of Science database.

**Figure 3 healthcare-10-00075-f003:**
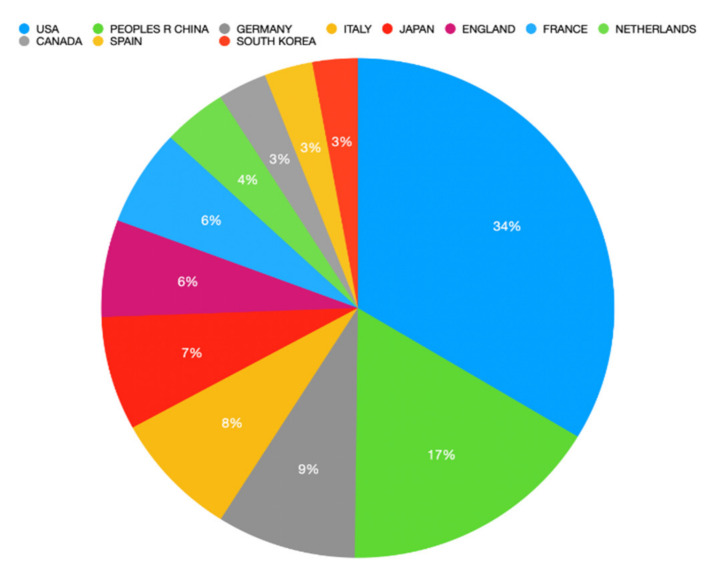
Top 10 countries regarding the number of publications related to immunotherapy research and colorectal cancer.

**Figure 4 healthcare-10-00075-f004:**
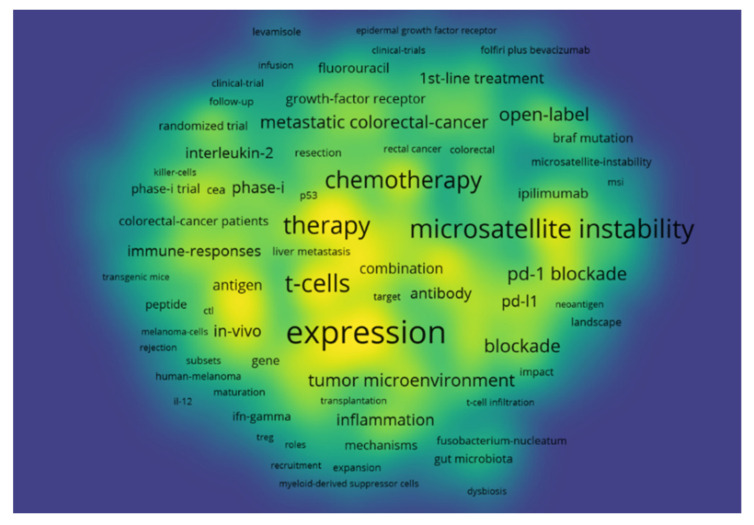
Bibliometric analysis of the keywords in the publications of immunotherapy in CRC showing the co-occurrence of keywords, where the size of nodes indicates the frequency of occurrence.

**Figure 5 healthcare-10-00075-f005:**
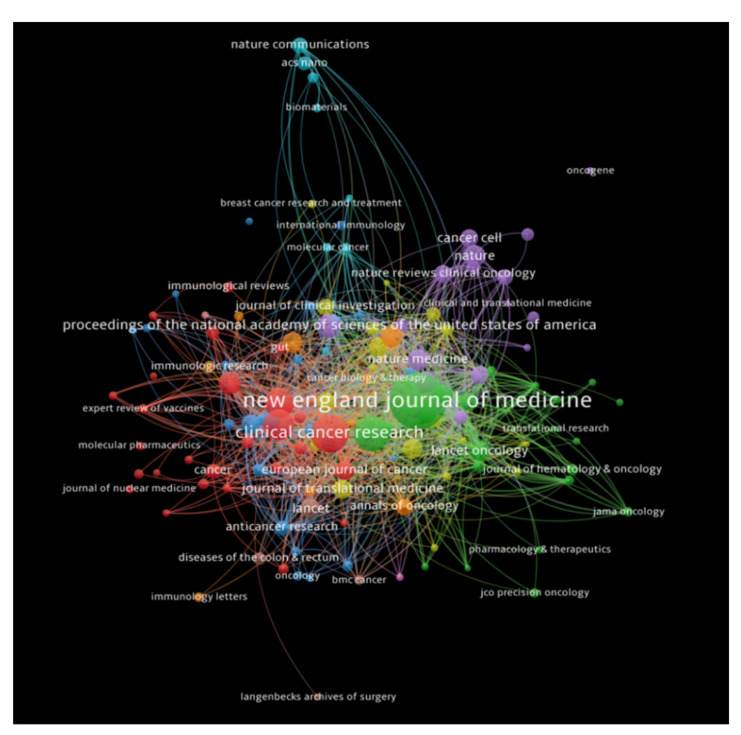
Bibliometric analysis of the journals in the publications of immunotherapy in CRC showing journals receiving top citations, where the size of nodes indicates the number of citations received.

**Figure 6 healthcare-10-00075-f006:**
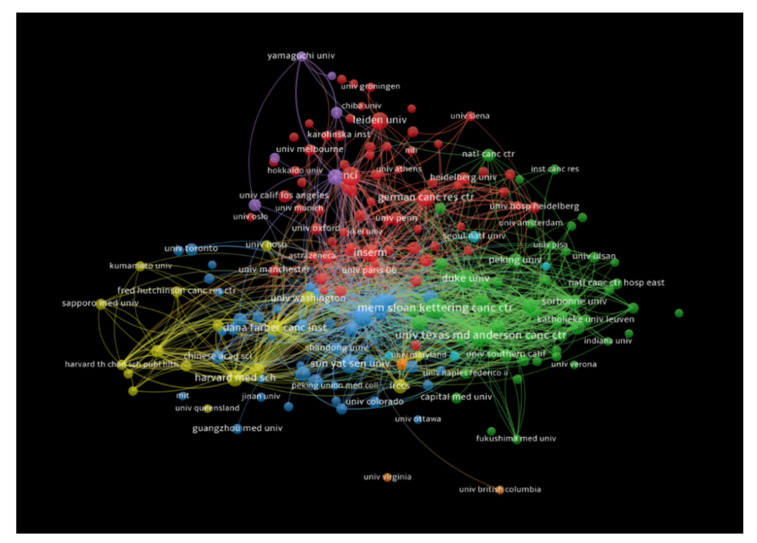
Bibliometric analysis of the institutions in the publications of immunotherapy in CRC showing institutions receiving top citations, where the size of nodes indicates the number of citations received.

**Figure 7 healthcare-10-00075-f007:**
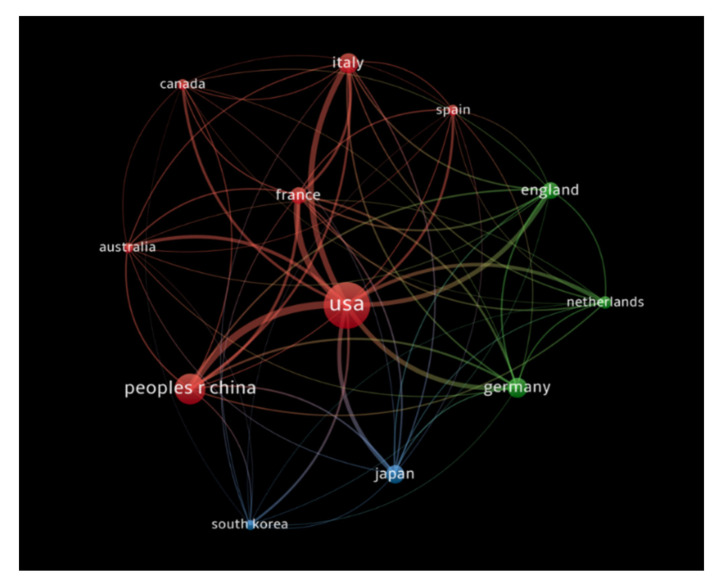
Bibliometric analysis of the countries in the publications of immunotherapy in CRC showing countries receiving top citations, where the size of nodes indicates the number of citations received.

**Figure 8 healthcare-10-00075-f008:**
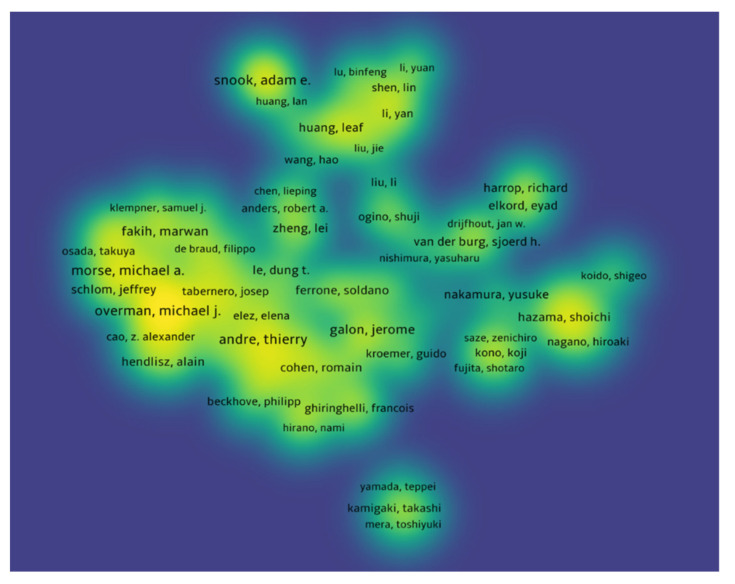
Bibliometric analysis of the authors of publications of immunotherapy in CRC showing authors receiving top citations, where the density indicates the number of citations received.

**Table 1 healthcare-10-00075-t001:** Top 10 institutions in the field immunotherapy research related to colorectal cancer.

Institution	Number of Articles	Percentage of the Total
Institut National De La Sante Et De La Recherche Medicale Inserm	144	3.621%
University of Texas System	140	3.520%
Harvard University	119	2.992%
Utmd Anderson Cancer Center	113	2.841%
Assistance Publique Hopitaux Paris Aphp	108	2.716%
Sorbonne Universite	103	2.590%
Memorial Sloan Kettering Cancer Center	100	2.514%
Unicancer	97	2.439%
Helmholtz Association	95	2.389%
Johns Hopkins University	94	2.364%
National Institutes Of Health Nih Usa	91	2.288%
German Cancer Research Center Dkfz	86	2.162%

**Table 2 healthcare-10-00075-t002:** Top 10 journals regarding the number of papers published related to immunotherapy in colorectal cancer according to the Web of Science database.

Journal	Impact Factor	Publisher	Number of Publications	Percentage
*Journal of Clinical Oncology*	32.96	Amer. Soc. Clinical Oncology	146	3.671%
*Cancer Immunology Immunotherapy*	5.44	Springer	117	2.942%
*Oncoimmunology*	5.87	Landes Bioscience	95	2.389%
*Cancer Research*	9.73	Amer. Assoc. Cancer Research	91	2.288%
*Clinical Cancer Research*	10.11	Amer. Assoc. Cancer Research	90	2.263%
*Cancers*	6.13	Mdpi	73	1.836%
*Annals of Oncology*	18.27	Oxford Univ. Press	72	1.810%
*Journal For Immunotherapy of Cancer*	10.25	BMC	67	1.685%
*International Journal of Cancer*	5.15	Wiley	62	1.559%
*Frontiers In Immunology*	5.09	Frontiers Media SA	60	1.509%

**Table 3 healthcare-10-00075-t003:** Top 10 highly cited papers related to immunotherapy in colorectal cancer according to the Web of Science database.

Rank	Title	Citation Count	Journal	Document Type	Publisher	Open Access	Year Published
1	Safety, Activity, and Immune Correlates of Anti-PD-1 Antibody in Cancer [12]	7724	*New England Journal of Medicine*	Article	Massachusetts Medical Soc.	Yes	2012
2	Safety and Activity of Anti-PD-L1 Antibody in Patients with Advanced Cancer [13]	4924	*New England Journal of Medicine*	Article	Massachusetts Medical Soc.	Yes	2012
3	PD-1 Blockade in Tumors with Mismatch-Repair Deficiency [4]	4454	*New England Journal of Medicine*	Article	Massachusetts Medical Soc.	Yes	2015
4	Phase I Study of Single-Agent Anti-Programmed Death-1 (MDX-1106) in Refractory Solid Tumors: Safety, Clinical Activity, Pharmacodynamics, and Immunologic Correlates [14]	1934	*Journal of Clinical Oncology*	Article	Amer. Soc. Clinical Oncology	Yes	2010
5	Intraepithelial CD8(+) Tumor-Infiltrating Lymphocytes and A High CD8(+)/Regulatory T Cell Ratio Are Associated with Favorable Prognosis in Ovarian Cancer [15]	1560	*Proceedings of the National Academy of Sciences of The United States of America*	Article	Natl. Acad. Sciences	Yes	2005
6	Association Of PD-1, PD-1 Ligands, and Other Features of the Tumor Immune Microenvironment with Response to Anti-PD-1 Therapy [16]	1472	*Clinical Cancer Research*	Article	Amer. Assoc. Cancer Research	Yes	2014
7	Effector Memory T Cells, Early Metastasis, and Survival in Colorectal Cancer [17]	1418	*New England Journal of Medicine*	Article	Massachusetts Medical Soc.	Yes	2005
8	Understanding The Tumor Immune Microenvironment (TIME) for Effective Therapy [18]	950	*Nature Medicine*	Review	Nature Publishing Group	Yes	2018
9	Cetuximab-Induced Anaphylaxis and Ige Specific for Galactose-Alpha-1,3-Galactose [19]	892	*New England Journal of Medicine*	Article	Massachusetts Medical Soc.	Yes	2008
10	Normalization of the Vasculature for Treatment of Cancer and Other Diseases [20]	897	*Physiological Reviews*	Review	Amer. Physiological Soc.	Yes	2011

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
