# Peer review of "Immunotherapy-Related Publications in Colorectal Cancer: A Bibliometric Analysis"

_healthcare, 2021, doi:10.3390/healthcare10010075_

Round 1

Reviewer 1 Report

Dear Authors

please find a minor changes in the attached file

Author Response

Manuscript ID: healthcare-1488035

Title: “Immunotherapy-related Publications in Colorectal Cancer: a Bibliometric Analysis ” by Sur et al.

Reviewer 1 comments:

Dear Authors

Please find a minor changes in the attached file.

We want to thank the reviewer for reviewing our manuscript. We are confident that by answering to the reviewer’s requests we will improve the current paper. Our main purpose is to present clear data which can be helpful for the clinicians and researchers involved in the immunotherapy field of gastrointestinal tumors.

We have amended our manuscript according to the reviewer’s suggestions. The changes were done using track changes.

Thank you for taking your time and analyzing our manuscript. Merry Christmas.

Reviewer 2 Report

Healthcare journal

Dear Editor in chief,

I have been asked to review the manuscript, “Immunotherapy-related Publications in Colorectal Cancer: A Bibliometric Analysis” by Sur et al. (healthcare-1488035). The authors aimed to assess re-search trends in CRC immunotherapy. Publication patterns of articles covering immunotherapies in CRC in the Web of Science Core Collection database were retrospectively examined using VOS viewer software (version 1.6.16) prior to May 25th, 2021.

The article is written in good English. However, the main objective of the study is scientometric evaluation of the publications in the field of immunotherapy, which are published in web of science. There are major issues, which should be addressed thoroughly or revised in the article. Otherwise, it is not suitable for publication in Healthcare journal.

1- As a systematic review, the authors should have addressed more than 3 (optimum 5 major databases including Pubmed, Scopus, Medline, EBSCO and also grey literature). Moreover, the process of finalizing search syntax, and inclusion of keywords is not well defined.

2- The search protocol should have been registered in PROSPERO or published (A big suggestion).

3- The quality of included articles are not assessed. (A major issue).

4- The discussion is majorly based on the statistics of published articles. (How many articles published, how many citations and …). However, what the readers expect is to comprehend an overview of the Immunotherapy-related approaches.

Author Response

Manuscript ID: healthcare-1488035

Title: “Immunotherapy-related Publications in Colorectal Cancer: a Bibliometric Analysis” by Sur et al.

Reviewer 2 comments:

I have been asked to review the manuscript, “Immunotherapy-related Publications in Colorectal Cancer: A Bibliometric Analysis” by Sur et al. (healthcare-1488035). The authors aimed to assess re-search trends in CRC immunotherapy. Publication patterns of articles covering immunotherapies in CRC in the Web of Science Core Collection database were retrospectively examined using VOS viewer software (version 1.6.16) prior to May 25th, 2021.

The article is written in good English. However, the main objective of the study is scientometric evaluation of the publications in the field of immunotherapy, which are published in web of science. There are major issues, which should be addressed thoroughly or revised in the article. Otherwise, it is not suitable for publication in Healthcare journal.

We want to thank the reviewer for the time spent reviewing our manuscript. We are confident that by answering to the reviewer’s requests we will improve the current paper. Our main purpose is to present clear data which can be helpful for the clinicians and researchers to find hot spots and collaboration opportunities in the field.

  • As a systematic review, the authors should have addressed more than 3 (optimum 5 major databases including Pubmed, Scopus, Medline, EBSCO and also grey literature). Moreover, the process of finalizing search syntax, and inclusion of keywords is not well defined.
  1. We appreciate the comment of the reviewer. We underline the fact that this is not a systematic review. The current paper is a bibliometric analysis of the literature that doesn’t follow the principles of a systematic review suggested by Cochrane Handbook of Systematic Reviews and Meta-Analysis. Being a bibliometric analysis this has the purpose of finding valuable literature, analyzing citation aspects and major institutions that have research in the employed field. According to Yale University Library a bibliometric analysis measures the research impact in a field (https://guides.library.yale.edu/impact/tools).

  • The search protocol should have been registered in PROSPERO or published (A big suggestion).
  1. We thank the reviewer for the comment. As we are clinicians involved in systematic reviews and meta-analysis and we follow the basic training provided by the course “Introduction in Systematic Reviews and Meta-Analysis” from John Hopkins University and also Cochrane members we register all our SR and MA to PROSPERO. Considering the aforementioned aspects, it’s not necessary to register to PROSPERO a bibliometric or scientometric analysis. Due to the fact that for the moment PROSPERO (https://www.crd.york.ac.uk/prospero/) handles mainly COVID-19 systematic reviews and meta-analysis as priority it is advised to register only systematic reviews and meta-analysis. We highly recommend that all the authors of systematic reviews and meta-analysis register as soon as possible their research and we fully support PROSPERO. As a matter of fact, our collaborators from Georgetown University (USA) from the Epidemiology Department suggest that only bibliometrics analysis of systematic reviews could be considered for registration to PROSPERO mentioning that this is not necessary.
  • The quality of included articles are not assessed. (A major issue).
  1. We want to thank the reviewer for the comment. According to the protocols of bibliometric analysis it is not necessary to do an evaluation of the quality of the papers. Moreover, at the moment there isn’t any standard evaluation grading for bibliometrics. As it is different from a systematic review, quality evaluation is not mandatory.

4- The discussion is majorly based on the statistics of published articles. (How many articles published, how many citations and …). However, what the readers expect is to comprehend an overview of the Immunotherapy-related approaches.

  1. We want to thank the reviewer for the suggestion of improving our manuscript. We have amended the discussion section according to your input. Now the phrase can be read like: “ Health authorities are increasingly recognizing the role of bibliometrics in evidence- based policy and patient care. In every country, knowledge generation in oncology is a vital phase in the care process. By leading and supporting government cancer control strategies, this can have a substantial impact on patient outcome”.

“To the best of our knowledge, this is the first bibliometric investigation of immunotherapy in colorectal cancer. The vast majority of publications in the subject of immunotherapy in CRC were covered by the data obtained from the available sources. The data presented might not represent the entire picture of the immunotherapy research in CRC, because conference papers, medical thesis and patents were not included in the document screening. Furthermore, bibliometrics indicators have a number of limitations and caution should be used when interpreting the results of the study.”

 Thank you for taking your time and analyzing our manuscript. Merry Christmas.

Reviewer 3 Report

In this manuscript Sur et al, analysed the research trend of immunotherapy to CRC focusing their attention to the past 10 years.

The work is well conducted and several parameters have been analysed. However, I do not see how, some of the parameters analysed, could influence and be useful to the current research on CRC. For example, how the co-authorship parameter can influence current research on CRC?

I would have like to see more discussion about the novelty of the paper and how this manuscript can help researcher in the filed to improve their own projects on CRC.

Therefore, I would suggest the author to revise the discussion of the paper and give an idea on how their manuscript can provide a reference for ongoing research in the field of immunotherapy and CRC. A better reeasoning on why some of the parameters have been anlysed would be useful and well seen. Otherwise I see this as a pure description of material and methods.

Author Response

Manuscript ID: healthcare-1488035

Title: “Immunotherapy-related Publications in Colorectal Cancer: a Bibliometric Analysis” by Sur et al.

Reviewer 3 comments:

In this manuscript Sur et al, analysed the research trend of immunotherapy to CRC focusing their attention to the past 10 years.

The work is well conducted and several parameters have been analysed. However, I do not see how, some of the parameters analysed, could influence and be useful to the current research on CRC. For example, how the co-authorship parameter can influence current research on CRC?

We want to thank the reviewer for the time spent reviewing our manuscript. We are confident that by answering to the reviewer’s requests we will improve the current paper. Our main purpose is to present clear data which can be helpful for the clinicians and researchers to find hot spots and collaboration opportunities in the field.

As bibliometrics analysis is a tool for evaluating research impact and also a measurement indicator for all aspects related to publications, co-authorship is a standard procedure in bibliometrics and scientometrics. These evaluations can have a major impact of evaluating international collaborations precluding collaborators from different countries, continents and backgrounds. Another important aspect of evaluating co-authorship is to see if authors from low or middle income have been implicated in high end research. This points out the hotspots in different regions for research in immunotherapy. Having this in mind, researchers in the field of gastrointestinal immunotherapy can find new ways of improving their research by reaching out to their counterparts from other regions and plan joint research ventures.

I would have like to see more discussion about the novelty of the paper and how this manuscript can help researcher in the filed to improve their own projects on CRC.

We want to thank the reviewer for the suggestion of improving our manuscript. We have amended the discussion section according to your input. Now the phrase can be read like: “To the best of our knowledge, this is the first bibliometric investigation of immunotherapy in colorectal cancer. The vast majority of publications in the subject of immunotherapy in CRC were covered by the data obtained from the available sources. The data presented might not represent the entire picture of the immunotherapy research in CRC, because conference papers, medical thesis and patents were not included in the document screening. Furthermore, bibliometrics indicators have a number of limitations and caution should be used when interpreting the results of the study.”

Therefore, I would suggest the author to revise the discussion of the paper and give an idea on how their manuscript can provide a reference for ongoing research in the field of immunotherapy and CRC. A better reeasoning on why some of the parameters have been anlysed would be useful and well seen. Otherwise I see this as a pure description of material and methods.

We want to thank the reviewer for the suggestion. We have amended the manuscript according to your input to underline the importance of continuous research in immunotherapy field. Now the phrase can be read like: “Health authorities are increasingly recognizing the role of bibliometrics in evidence-based policy and patient care. In every country, knowledge generation in oncology is a vital phase in the care process. By leading and supporting government cancer control strategies, this can have a substantial impact on patient outcome. “

The parameters that are analyzed are according to the general guidelines of bibliometrics and scientometrics.

(Haustein, S., Larivi`ere, V., 2015. The use of bibliometrics for assessing research: possibilities, limitations and adverse effects. In: Incentives and Performance. Springer, pp. 121–139.)

(Kokol, P., Blazun Vosner, H., Zavrsnik, J., 2020. Application of bibliometrics in medicine: a historical bibliometrics analysis. Health Info. Libr. J. https://doi.org/10.1111/ hir.12295. 10.1111/hir.12295.)

We thank the reviewer for the comment and we understand their concern about the technicality of the research, but we assure the reviewer that we encompassed every correct step of bibliometrics analysis to make sure that the readers have the most important and up-to-date information. We agree that bibliometrics is a very technical method to analyze the structure of literature, but can considerably impact research projects, guide the design of future studies, and boost the contribution in scientific productivity. Also, Bibliometrics plays an important role in the assessment of landscapes of particular research fields (Belter, C.W., 2015. Bibliometric indicators: opportunities and limits. J. Med. Libr. Assoc. 103 (4), 219–221.).

Thank you for taking your time and analyzing our manuscript. Merry Christmas.

Round 2

Reviewer 2 Report

The revised version is appropriate.

Author Response

Thank you for the suggestions that allow us to improve our study. 

Reviewer 3 Report

I was asked to review the manuscript: "Immunotherapy-related Publications in Colorectal Cancer: a Bibliometric Analysis" from Sur et al, after the first round of revision.

The authors answered to the different concerns raised and I am in favor now for publications.

Minor comments:

1) there are some typos to correct

2) I find that the sentence that has been added at page 2 in the introduction is not necessary and it does not bring anything more to the text. It can be removed.

Thanks for your work.

Author Response

Dear Reviewer, 

1.) We double checked the spelling and corrected the minor typos.

2.) We deleted the sentence at page 2. 

Thank you for the careful analysis of our article.